# Shape Memory Polymer Foams Synthesized Using Glycerol and Hexanetriol for Enhanced Degradation Resistance

**DOI:** 10.3390/polym12102290

**Published:** 2020-10-06

**Authors:** Sayyeda Marziya Hasan, Grace K. Fletcher, Mary Beth Browning Monroe, Mark A. Wierzbicki, Landon D. Nash, Duncan J. Maitland

**Affiliations:** 1Shape Memory Medical Inc., Santa Clara, CA 95054, USA; landon@shapemem.com; 2Department of Biomedical Engineering, Texas A&M University—Main Campus, College Station, TX 77843, USA; gracekfletcher@gmail.com (G.K.F.); mbbmonroe@gmail.com (M.B.B.M.); mwierzbicki03@gmail.com (M.A.W.)

**Keywords:** shape memory polymer, degradation-resistant, polyurethane foam

## Abstract

Shape memory polymer foams have been used in a wide range of medical applications, including, but not limited to, vessel occlusion and aneurysm treatment. This unique polymer system has been proven to shape-fill a void, which makes it useful for occlusion applications. While the shape memory polymer foam has superior performance and healing outcomes compared to its leading competitors, some device applications may benefit from longer material degradation times, or degradation-resistant formulations with increased fibrous encapsulation. In this study, biostable shape memory polymer foams were synthesized, and their physical and chemical properties were characterized as an initial evaluation of feasibility for vascular occlusion applications. After characterizing their shape memory behavior in an aqueous environment, degradation of this polymer system was studied in vitro using accelerated oxidative and hydrolytic solutions. Results indicated that the foams did not lose mass under oxidative or hydrolytic conditions, and they maintained high shape recovery in aqueous in vitro models. These degradation-resistant systems have potential for use in vascular occlusion and other wound healing applications that benefit from permanent, space-filling shape memory behavior.

## 1. Introduction

Shape memory polymers (SMPs) have been studied extensively in efforts to utilize their unique shape changing properties in a wide range of commercial applications such as aerospace, textiles, robotics, and medicine [1,2]. Shape memory polymers are stimulus-responsive systems that have the capability to transition between two shapes [1,3,4]. An external stimulus, such as heat, may be applied to the polymer, causing it to reach its rubbery transition. During application of heat, the polymer can be molded into a secondary shape and cooled back to its glassy state. This cooling process will cause the polymer to hold its programmed shape until heat is re-applied to actuate it back to its original shape. 

SMPs are becoming more widespread for use in medical applications. Historically, they have been used in “smart” sutures and stents [5], but more recently their applications have extended into the vascular occlusion space [6,7,8]. Lendlein et al. created a thermally-activated SMP with oligo(ε-caprolactone) diol and oligo(p-dioxanone) to develop self-closing sutures [9]. Upon increasing the temperature above the polymer’s transition temperature, the polymer strands would tighten due to the suture shrinking as a direct response to a heat stimulus. Wong et al. developed a radiopaque SMP embolization plug for temporary occlusion of the hepatic artery suitable for treatment of hepatocellular carcinomas [10]. This heat-activated plug was designed to block arterial flow, therefore halting blood supply to a cancerous tumor. Similarly, Boyle et al. reported on a SMP foam-coated coil that demonstrated effectiveness in occluding an in vitro saccular aneurysm model [8]. Aneurysms are ballooning in a blood vessel due to weakening of the vessel wall [8]. If left untreated, this abnormality can rupture and cause internal hemorrhaging. The foam-coated coil, designed by Boyle et al. would ideally fill the aneurysm sac with a self-expanding foam that would result in clot formation and ultimately seal off the aneurysm from the parent vessel. 

Another application of smart polymers in medical spaces is the polyurethane SMP foam-based plug, designed by Landsman et al. for its application in treating chronic venous insufficiency and varicoceles [11]. In this application, the crimped foam can be delivered to the defect site through a catheter, after which it actuates and expands to its primary shape when introduced to the aqueous in vitro environment, causing the polymer to occlude the blood vessel.

The SMP foam system has been researched extensively by the Maitland group for use in biomedical devices. This material system is an amorphous, shape memory polymer with high porosity and ultra-low density [3], Additionally, physical and chemical properties of the material can be tuned depending on the final application [12,13,14]. Singhal et al. modified the actuation rate of the SMP foam by altering its chemical structure. This research showed that changes in the polymer hydrophobicity and glass transition temperature (T_g_) can be used to delay foam actuation. Hasan et al. incorporated radiopaque nanofillers into the polymer matrix to introduce x-ray visibility to the foams [15]. The SMP foam is also biocompatible and provides a superior healing outcome compared to commercial occlusion products, such as bare metal coils [16]. 

Material properties of a polymer system are a critical determinant of its use in medical or commercial applications. One property that heavily influences material performance in vivo is biostability. Biodegradation of a polymer system can occur via four degradation mechanisms: hydrolysis, oxidation, enzymatic degradation, and physical degradation [17] Unintended biodegradation can result in device breakdown which may lead to further complications in the patient over time. An example of unexpected material degradation was the breakdown of a polyurethane pacemaker lead coating that resulted in electrical disfunction [18].

The biodegradation of the SMP foam, created by the Maitland group, was studied by Weems et al. to evaluate its degradation rate under hydrolytic and oxidative conditions representative of an in vivo environment [19]. The study concluded that while the foams maintained hydrolytic stability, they were susceptible to oxidative degradation. Namely, the foams exhibited complete mass loss within ~15 days in accelerated oxidative conditions (50% H_2_O_2_) [19]. The original polymer network is predominantly polyurethane/urea, which is synthesized by reacting multifunctional hydroxyl-terminated monomers with linear diisocyanates. The hydroxyl components in this network contains tertiary amine groups, which are susceptible to oxidation, generating ammonia and carboxylic acid as terminal by-products. In order to maintain prolonged mechanical strength and minimize mass loss after implantation, this work explores new, degradation-resistant foam compositions. 

To that end, novel SMP foams were developed to exhibit oxidative and hydrolytic stability while maintaining the shape memory behavior that is desirable for long-term stimulus-responsive implants. The chemical structure of the polymer network was modified to remove the tertiary amine groups that were the key points of chain cleavage in the original design. Thermal and chemical properties of the resulting polymer system were studied, and degradation profiles were evaluated under hydrolytic and oxidative conditions.

## 2. Materials and Methods

### 2.1. Synthesis and Post-Processing

1,2,6-hexanetriol (HT, 96%, Alfa Aesar, Haverhill, MA, USA), glycerol (G, ≥99.5%, Sigma-Aldrich, St. Louis, MO, USA), 2-butyl-2-ethyl-1,3-propanediol (BEP, >98% TCI America Inc., Portland, OR, USA), hexamethylene diisocyanate (HDI, TCI America Inc., Portland, OR, USA), Vorasurf DC 198 (Dow Corning, Midland, MI, USA), Vorasurf DC 5943 (Dow Corning, Midland, MI, USA), DABCO T-131 (Evonik Industries AG, Essen, Germany), DABCO BL-22 (Evonik Industries AG, Essen, Germany), Enovate 245fa Blowing Agent (Honeywell International, Inc., Charlotte, NC, USA) deionized (DI) water (ASTM Type II; LabChem, <1 µS/cm), HPLC Grade Water (PHARMCO, 0.5 µS/cm, Greenfield Global USA Inc., Brookfield, CT, USA), reverse osmosis water (RO Water, Texas A&M University, College Station, TX, USA) and isopropyl alcohol (IPA, ≥99.5%; Fisher Scientific, Waltham, MA, USA) were used as received.

SMP foams were synthesized using a gas-blowing process described by Hasan et al. [12]. To synthesize glycerol foams, first, an isocyanate (NCO) prepolymer was synthesized with appropriate molar ratios of G, BEP, and HDI. Next, a hydroxyl (OH) mixture was blended with the remaining molar equivalents of G and BEP, along with foaming additives such as DI water (<1 µS/cm), and catalysts. The NCO prepolymer and the OH mixture were combined in the foam cup along with surfactants, DC 198 and DC 5943, and Enovate. This mixture was mixed in a FlackTek speedmixer (FlackTek, Inc., Landrum, SC, USA) and poured in the foam bucket. The foam was allowed to cure fully before further processing. To synthesize hexanetriol foams, the abovementioned process was utilized with glycerol (G) being replaced by hexanetriol (HT). Table 1 shows the weight percent of each component used for foam synthesis and Figure 1 shows the corresponding molecular structure of each monomer and a schematic of the foaming process. 

Post-synthesis, the bulk foam was cut into 2 cm long cylinders with 6 mm diameter using a biopsy punch. The samples were annealed at 90 °C for 30 min and allowed to cool down completely before further processing. Foam cylinders were cleaned in 32 oz glass jars using one 30 min sonication wash in DI water and four 30 min sonication washes in IPA. After each wash, the solvent was discarded and the jars were replenished with fresh solvent. Prior to testing, foam cylinders were dried at 100 °C under vacuum for at least 12 h after which they were stored in a plastic storage container with dessicant. 

### 2.2. Chemical and Physical Characterization

#### 2.2.1. Cell Structure and Pore Size

Foam cubes were cut from the top, middle, and bottom sections of the foams using a resistive hot-wire cutter for pore size characterization. Sample preparation and instrumentation for pore size analysis were determined using ASTM D3576 [20]. Pore size was measured using the Keyence Inspection System (VHX-5000) with the variable angle adapter lens via 2-point linear measurements. Pore images were collected from samples that were axial (parallel) and transverse (perpendicular) to the foam rise, and average pore size values were calculated for each foam (10 points measured per image).

#### 2.2.2. Fourier-Transform Infrared (FTIR) Spectroscopy

Thin foam samples were cut (2–3 mm) from bulk foams, and FTIR spectra was collected using a Bruker ALPHA Infrared Spectrometer (Bruker, Billerica, MA, USA) via the Platinum ATR Sampling Module. Sixty-four background scans of the empty chamber were taken followed by 32 sample scans of the various foam compositions. FTIR spectra was collected in absorption mode at a resolution of 4 cm^−1^ within the wavenumber range of 400 to 4000 cm^−1^ with atmospheric compensation. OPUS software (Bruker, Billerica, MA, USA) was utilized to subtract the background scans from the spectra and to conduct a baseline correction for IR beam scattering and an atmospheric compensation to remove any peaks acquired due to carbon dioxide or water in the air. 

#### 2.2.3. Differential Scanning Calorimetry (DSC)

Foam samples (4–5 mg) were cut from bulk foam and thermally characterized using the Q-2000 DSC (TA Instruments, Inc., New Castle, DE, USA). The first cycle consisted of decreasing the temperature to −40 °C at 30 °C min^−1^ and holding it isothermally for 2 min. The temperature was then increased to 120 °C at 30 °C min^−1^ and held isothermally for 2 min. In the second cycle, the temperature was reduced to −40 °C at 30 °C min^−1^, held isothermally for 2 min, and raised to 120 °C at 10 °C min^−1^. T_g_ (dry) was recorded from the second cycle based on the inflection point of the thermal transition curve using TA instruments software. The aluminum pan was vented during this test to remove moisture from the sample during the first cycle. N = 3 was utilized per foam composition.

#### 2.2.4. Water Moisture Plasticized Glass Transition Temperature

Foam samples (4–5 mg) were cut from bulk foam and submerged in RO water at 50 °C for 5 min to allow full plasticization. After the samples were removed from water, they were pressed dry with Kim Wipes (Kimberly-Clark Professionals, Roswell, GA, USA), weighed, and placed in an aluminum pan sealed with a vented aluminum lid. A Q-2000 DSC was used to cool the samples to −40 °C, hold them isothermally for 2 min, and heat them to 80 °C at 10 °C·min^−1^. TA instruments software was used to generate the thermogram and acquire the T_g_ (wet), after water plasticization, using the inflection point of the thermal transition. N = 3 was utilized per foam composition. 

#### 2.2.5. Thermogravimetric Analysis

Thermal stability of the SMP foams was determined using thermogravimetric analysis (TGA). Samples (10–15 mg, n = 1) were prepared from the bulk foam. A platinum pan was used to hold each sample and tared before each run. The samples were heated from 25 to 800 °C at 10 °C·min^−1^ under nitrogen flow of 60 mL·min^−1^ using a TGA Q 50 (TA Instruments, New Castle, DE, USA). At 800 °C, the gas was switched to air flow at 60 mL·min^−1^, and the samples were heated to 1000 °C. The thermograms were evaluated using TA Universal Analysis software, and percent mass remaining (%) versus temperature (°C) curves were graphed for each foam composition.

#### 2.2.6. Mechanical Testing

Mechanical properties of the SMP foams were determined using MTS Criterion Model 42 (MTS Systems Corporation, Inc., Eden Prairie, MN, USA), A 10 N load cell was used for conducting tensile (extension) testing with a test speed of 5 mm/min. N = 10 samples were prepared for each SMP foam formulation. Then, 2 mm thick foam slices were cut from the bulk foam using the Proxxon 37080 Hot Wire Cutter (PROXXON Inc., Hickory, NC, USA). The foam slices were further cut into dog bones using a laser engraving system (Orion Motor Tech, Lake Forest, CA, USA). The foam dog bones were endcapped with wooden blocks to prevent material damage during clamping and annealed for 30 min at 90 °C. The samples were allowed to cool down to room temperature before testing. Five samples, for each formulation, were tensile tested under dry, ambient conditions and the remaining 5 samples were soaked in room temperature DI water for 48 h, to ensure complete water-plasticization, prior to tensile testing. Strain at break (%) and Young’s Modulus (kPa) was recorded for both the glycerol and the hexanetriol SMP foams under dry and water-plasticized conditions. 

#### 2.2.7. Shape Fixity and Shape Recovery

Foam cylinders (n = 3) with a diameter of 10 mm and height of 10 mm were cut out of the bulk foam for each SMP formulation. The samples were annealed for 30 min at 90 °C and allowed to cool to room temperature, prior to testing. Each sample, from the glycerol and hexanetriol formulation, was heated to 100 °C, using a heat gun, and programmed to a compressed height of 3 mm. Shape setting occurred by cooling the sample to room temperature while the sample remained under load. Once the sample was completely cool, the load was removed and the programmed height (*h*_1_) was measured using a microcaliper. The sample was allowed to remain in the compressed shape, without load, at room temperature, for 15 min after which the programmed height was remeasured (*h*_2_). After this step, the sample was reheated to 100 °C, using a heat gun, allowed to recover to its original shape without any load. The recovered height was measured using micro-calipers. Shape fixity (R_f_) and shape recovery (R_r_) was calculated for each sample using Equations (1) and (2), respectively.
(1)$$Shape\;Fixity=\left(\frac{h_{2}}{h_{1}}\right)\times 100$$
(2)$$Shape\;Recovery=\left(\frac{Recovered\;height}{Original\;height}\right)\times 100$$

### 2.3. In Vitro Shape Memory Behavior

#### Volume Recovery and Expansion

Cleaned foam cylinders (n = 3) were used to evaluate actuation profiles of each SMP foam composition. A 203.20 μm diameter nickel-titanium (Nitinol) wire (NDC, Fremont, CA, USA) was inserted through the center of the sample along its length to serve as a stabilizer. The foam samples were radially compressed to their smallest possible diameter using a Machine Solutions crimper-306630-103 (Machine Solutions, Flagstaff, AZ, USA) by heating the material to 100 °C, holding it isothermally for 15 min, and cooling the foams back down to program them to the crimped morphology. Initial foam diameter (5 measurements per sample) was measured and recorded for each sample using photos of the samples with a calibrated ruler and Image J software (NIH, Bethesda, MD, USA). The foams were placed in a water bath at 50 °C, removed after 20 min, and allowed to cool to room temperature. While in the heated water bath, videos of the crimped samples with a calibrated ruler were recorded to monitor time until full foam expansion was achieved. The final diameter of the samples (5 measurements per sample) was measured and recorded, using photos of the samples with a calibrated ruler and Image J software. Frames from multiple time points from the actuation videos were used to measure cylinder diameter (5 measurements per sample) and to graph foam diameter vs. time. Volume expansion was calculated using Equation (3), and volume recovery was calculated using Equation (4).
(3)$$Volume\;expansion={\left(\frac{\mathrm{Recovered}\;\mathrm{diameter}}{\mathrm{Compressed}\;\mathrm{diameter}}\right)}^{2}$$
(4)$$Volume\;recovery={\left(\frac{\mathrm{Recovered}\;\mathrm{diameter}}{\mathrm{Original}\;\mathrm{diameter}}\right)}^{2}\times 100\%$$

### 2.4. In Vitro Degradation

#### 2.4.1. Accelerated Hydrolytic Degradation

Cleaned foam cylinders (n = 5) were incubated individually in 0.1 M NaOH at 37 °C for 70 days. The dry samples were weighed at t = 0. Every 13–17 days, samples were washed with water and ethanol, dried under vacuum at 50 °C for 24 h, and then weighed. The degradation solutions were changed every 7 days. Average mass per time point was calculated, and percent mass remaining was plotted versus time. 

#### 2.4.2. Accelerated Oxidative Degradation

Cleaned foam cylinders (n = 5) were incubated individually in 50% H_2_O_2_ at 37 °C for 42 days. The samples were weighed at t = 0 and weighed again approximately every 3 days after washing in water and ethanol and drying under vacuum at 50 °C for 24 h. After each time point, the samples were incubated in fresh degradation solution. Average mass per time point was calculated and percent mass remaining was plotted versus time.

Statistical analysis was performed using a Student’s *T*-test of each time point’s masses in comparison to the initial masses at 0 days. *p* < 0.05 was taken to be statistically significant. These data are reported in Appendix A.

## 3. Results

### 3.1. Chemical and Physical Characterization

Cell structure and pore sizes are shown in Figure 2 and Table 1. The SMP foams have interconnected, anisotropic pores with partially open cell membranes. The difference in pore sizes can be attributed to the varying catalyst concentrations used per composition. Higher catalyst concentrations will generally yield a smaller pore size due to the polymer curing during small bubble formation. Glycerol foams had an axial pore size of 1404 ± 121 Cell structure and pore sizes are shown in Figure 2 and Table 2. The SMP foams have interconnected, anisotropic pores with partially open cell membranes. Glycerol foams had an axial pore size of 1404 ± 121 µm and a transverse pore size of 1107 ± 69 µm, while hexanetriol foams yielded smaller pores (axial: 957 ± 82 µm, transverse: 580 ± 47 µm). The difference in pore sizes between the two formulations can be attributed to the varying catalyst concentrations used per composition (i.e., a higher concentration of gelling catalyst was required for hexanetriol foam synthesis). Higher catalyst concentrations will generally yield a smaller pore size due to the polymer curing during small bubble formation. 

Additionally, variation in pore sizes was observed between the two foaming directions (axial vs. transverse) for each foam. Pore sizes were generally longer in the axial direction due to the foam curing while sticking against the walls of the foaming mold.

The FTIR spectra in Figure 3 show the presence of urethane and urea carbonyl peaks at 1689 and 1646 cm^−1^, respectively. The amine stretch at 3307 cm^−1^ is contributed from the amine functionality present in the urethane/urea linkages. These peaks confirmed the chemical nature of these materials to be predominantly polyurethane/urea. Urea generation occurred upon reaction of the chemical blowing agent, water, with free isocyanate groups to result in carbondioxide and primary amines. The resulting amines further reacted with isocyanate molecules to create urea functionalities. Additionally, the amide III at 1237 cm^−1^ and C–H peaks at 2919 and 2852 cm^−1^ are attributed to the backbone of the hydroxyl and isocyanate monomers. 

The glass transition temperatures (T_g_) of the two polymer systems are reported in Table 3 and displayed in Figure 4. The dry T_g_s of both formulations are well above room temperature at 54.2 ± 0.7 °C and 49.4 ± 0.3 °C for glycerol and hexanetriol foams, respectively. However, wet foams experienced a significant decrease in transition temperature at 15.6 ± 1.1 °C and 15.5 ± 0.5 °C for glycerol and hexanetriol foams, respectively. Figure 4 shows a representative comparison of wet and dry DSC traces for one sample per foam formulation to highlight the shifts in T_g_ onset and inflection as a direct effect of exposure to moisture

Thermal degradation thermograms, Figure 5, show that both SMP compositions maintain thermal stability until 200 °C, after which the polymer is degraded until no mass was remaining around 500 °C. The thermal stability of the polymers was confirmed by comparing the FTIR spectra before and after heating the materials to 150 °C for 30 min where no shift in the peaks was noted. This study indicates that the urethane and urea groups are stable under heat and the urethane linkages are irreversible. These thermally stable systems allow for effective material programming using heat while the polymer maintains mechanical integrity during processing.

Mechanical properties of both SMP foams, Table 4, show that both formulations had high strain at break (% elongation) showing that both formulations exhibit elastomeric properties. The glycerol foam had a percent elongation of 274% ± 41% while the hexanetriol foam had a percent elongation of 247% ± 29%. Young’s modulus of the glycerol foam and the hexanetriol foam was 130 ± 30 kPa and 390 ± 30 kPa, respectively. The higher modulus for the hexanetriol foam indicates that the foam had greater stiffness compared to the glycerol foam, which is also reflected by the lower percent elongation for this composition. Water-plasticized foam samples had higher percent elongation compared to the dry samples, for both formulations. Due to hydrogen-bonding disruption by water molecules, the polymer network was plasticized and thereby resulted in higher elongation. Percent elongation for water-plasticized glycerol foam was 558% ± 29% which is 2× the elongation of dry foam. Modulus values for the plasticized samples were 8 ± 1 kPa which is 16× less than the dry modulus, Percent elongation for water-plasticized hexanetriol foam was 521% ± 43% which is 2× the elongation of dry foam. Modulus value for the plasticized hexanetriol foam was 15 ± 3 kPa which is 26× less than the dry modulus. Both foam formulations experience similar trends of increased percent elongation and decreased modulus when plasticized with water. 

Shape fixity (R_f_) and shape recovery (R_r_) of both SMP foams were high, with values greater than 95%, as shown in Table 4. The glycerol foam had shape fixity of 99.2% ± 0.2% with a shape recovery of 99.3% ± 0.8%. The hexanetriol foam had similarly high shape fixity of 98.0.2% ± 1.5% and shape recovery of 98.3% ± 0.3%. These high shape fixity and recovery values indicate that both SMP foam formulations were able to maintain their programmed shapes and recover to their original shape with minimal loss in shape memory properties.

### 3.2. In Vitro Shape Memory Behavior

Glycerol-based foams experienced 84% volume recovery to the original foam diameter with 34-fold volume expansion compared to the compressed foam, Table 5. Hexanetriol-based foams, however, demonstrated a volume recovery of 105% with 42-fold volume expansion. These results suggest that changing the hydroxyl component plays a role in shape memory behavior of the polymer system. Actuation profiles of the two compositions are displayed in Figure 6. Both foam compositions begin to actuate at 15 min in 50 °C water, however, glycerol foams recovered to the original diameter at a faster rate compared to hexanetriol foams. Both foams achieved close to full shape recovery within 40 min in an aqueous environment. However, it must be noted that glycerol foams demonstrated a faster shape memory response with less overall volume recovery while hexanetriol foams had a slower actuation response with ultimately higher volume recovery. 

### 3.3. In Vitro Degradation

The accelerated hydrolytic and oxidative degradation plots in Figure 7 show no significant mass changes relative to the time 0 masses over the time periods studied. Both SMP foams maintained 95% or higher mass while exposed to various degradation solutions. While the hydrolytic degradation results match those of previous studies, wherein SMP foams were stable over 36 weeks of testing in 0.1 M NaOH [20], these results are highly significant when taken with previous oxidative testing results. Namely, SMP foams experienced 100% mass loss within ~15 days in 50% H_2_O_2_ [20]. Here, we carried out oxidative testing for ~3× longer, with no significant mass loss (Appendix A).

## 4. Discussion

Two formulations of shape memory polymer foams were developed with inherent resistance to hydrolysis and oxidation. In vitro degradation results in Figure 7 show that the polymer systems can withstand extreme concentrations of oxidative and hydrolytic solutions without experiencing mass loss. This resistance to degradation can be attributed to the absence of functional groups in the polymer network that are susceptible to hydrolysis and oxidation. The glycerol and hexanetriol SMP foams have viability for use in implantable applications, especially ones that require sustained presence of the implant without mass loss over time to prevent a loss of device integrity. These data confirm that modifying the polymer chemistry to remove functionalities susceptible to oxidation, such as tertiary amines, will result in degradation-resistant systems that can withstand extreme degradation conditions without experiencing mass loss.

While oxidative stability of the foam was of significance, it must be noted that the two foam formulations also displayed sufficient shape memory behavior similar to previously designed biodegradable systems [21]. Volume recovery and expansion were affected by the type of hydroxyl monomer used in the polymer network. For example, hexanetriol has a 3-carbon alkane functionality which allows for larger volume expansion and percent recovery compared to the much smaller glycerol counterpart. Longer carbon linkages in hexanetriol allowed for greater molecular rotation, thus allowing higher water plasticization which ultimately resulted in a larger overall volume recovery compared to glycerol foams. Similarly, both foam formulations presented high shape fixity and shape recovery (>95%) under dry conditions, indicating that both systems maintain shape memory properties similar to the SMPs studied by Weems et al. [21].

Water plasticization is a phenomenon best represented by the transition temperatures of the two foams formulations under dry and wet conditions. Under dry conditions, both foam compositions had a transition temperature closer to 50 °C; however, once exposed to moisture, the polymer transition temperature shifted closer to 15 °C. This T_g_ suppression is due to plasticization of the polymer chains by water molecules. Hydrogen bonding between the urethane/urea linkages is disrupted when water molecules penetrate the polymer network and in turn partake in hydrogen bonding with the urethane/urea linkages. This temporary lubrication of the polymer network causes a significant drop in thermal transitions, which ultimately aids shape recovery and polymer actuation in an aqueous environment where the liquid temperature is well below the T_g_ of the dry polymer. Moreover, percent elongation of the wet foam, for both formulations, was higher when the polymer network was plasticized compared to when the material was dry. This shift in thermal and mechanical properties, in aqueous conditions, is indicative of how these materials would actuate in vivo. 

According to Figure 6, hexanetriol foams experienced slower actuation despite having greater volume recovery and expansion compared to glycerol foams. Once hexanetriol foams were plasticized they experienced larger volume recovery; however, system hydrophobicity played an important role in the rate of recovery. Glycerol has fewer carbon atoms, which decreased its hydrophobicity compared to hexanetriol. With a reduced hydrophobicity, water plasticization and subsequent actuation of the glycerol foams was achieved more quickly in comparison to hexanetriol foams. It is important to note that the actuation profile for the glycerol foam plateaus after 30 min and at 40 min, the foam had achieved maximum recovery. However, the hexanetriol foam continued to expand at 35 min and started to plateau at 40 min. This 5-min difference in achieving maximum recovery is noteworthy because while it appears insignificant at 50 °C, at body temperature the recovery rate may be different over the course of several minutes or hours. A slow recovery profile, at body temperature, may be beneficial for slow drug release applications in practical terms.

Furthermore, these novel SMP systems have tunable pore morphology that can be modified depending on the desired application. Modifying viscosity of the NCO prepolymer during the synthesis step and/or the catalyst contents provide mechanisms for generating foams with tunable pores. Additionally, actuation times and shape recovery could be further tuned by modifying the ratio of glycerol and hexanetriol. These foams displayed slower actuation profiles (~40 min to full expansion) which may be beneficial for applications that require longer delivery times or slow expansion profiles, such as sustained drug release. The degradation-resistant polymer foams designed in this work are promising as they maintained shape recovery while remaining stable under corrosive and oxidizing environments. A detailed evaluated of the biostability of these materials must be conducted in future studies to evaluate the SMP foam degradation profile in solutions such as PBS, which are more representative of an in vivo environment. Furthermore, changes to shape memory properties and chemical structure must also be evaluated on materials subjected to the aforementioned biostability studies. Similarly, enzymatic and physical degradation rates of these systems must be further evaluated along with chronic material compatibility in vivo. 

These non-degradable polymer systems have utility not only in vascular applications, but also as anchoring cannulas and surgical fasteners. Bettuchi and Heinrich developed a thermo-responsive surgical fastener that can be used to fix an implant, such as a hernia mesh, into place [22]. The shape memory polymers presented in this work was synthesized as foam, however the polymer formulation can also be cured neat for non-porous applications. A similar application for our degradation-resistant material system may be anchoring cannulas, in which the SMP is used to expand and secure the cannula in tissue after implantation [23]. While SMP systems have been studied extensively and have been used for developing a large variety of medical devices, the degradation-resistant SMPs developed in this work serve as potentially valuable additions to the database of polymer systems to consider for a given medical application. Therefore, additional studies should be conducted on this newly designed biostable foam to evaluate its biostability, biocompatibility in terms of cell viability, cell infiltration and attachment, and its shape memory behavior in vitro.

## Figures and Tables

**Figure 1 polymers-12-02290-f001:**
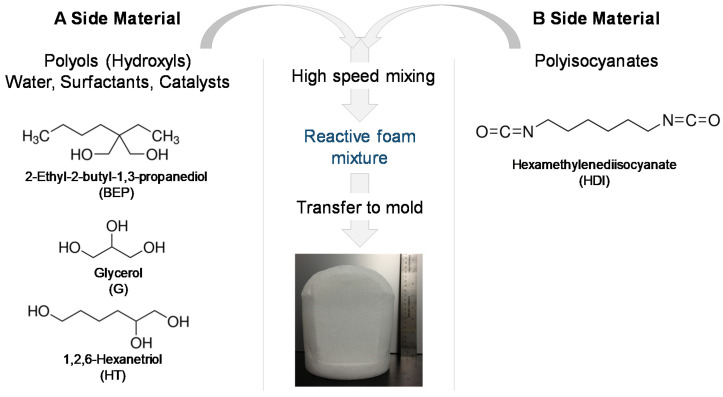
Molecular structure of monomers and steps used during the foaming process.

**Figure 2 polymers-12-02290-f002:**
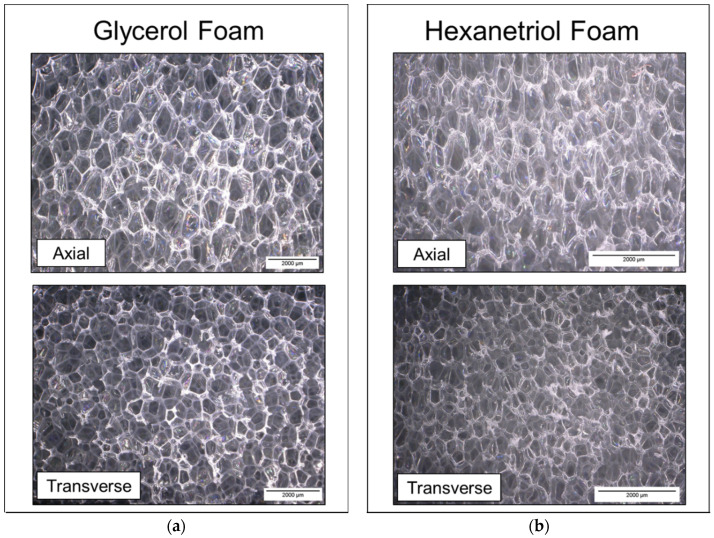
Axial and transverse pore morphology of glycerol foam (**a**) and hexanetriol foam (**b**).

**Figure 3 polymers-12-02290-f003:**
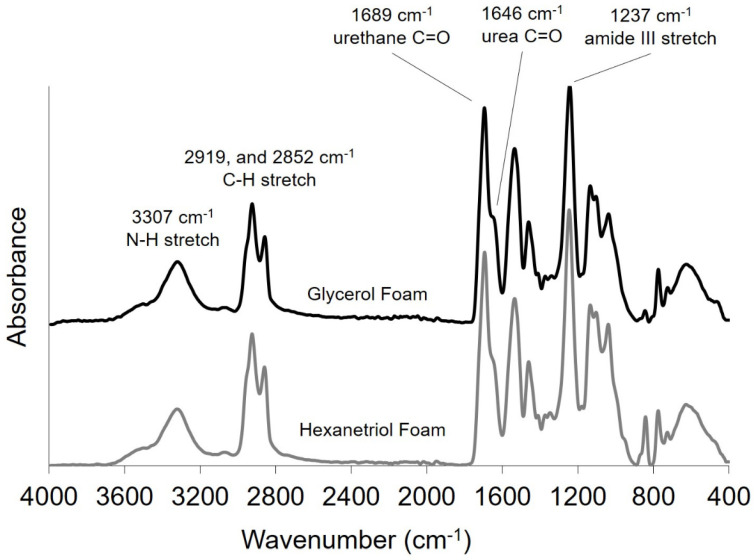
FTIR spectra of glycerol (top) and hexanetriol (bottom) SMP polymer foams.

**Figure 4 polymers-12-02290-f004:**
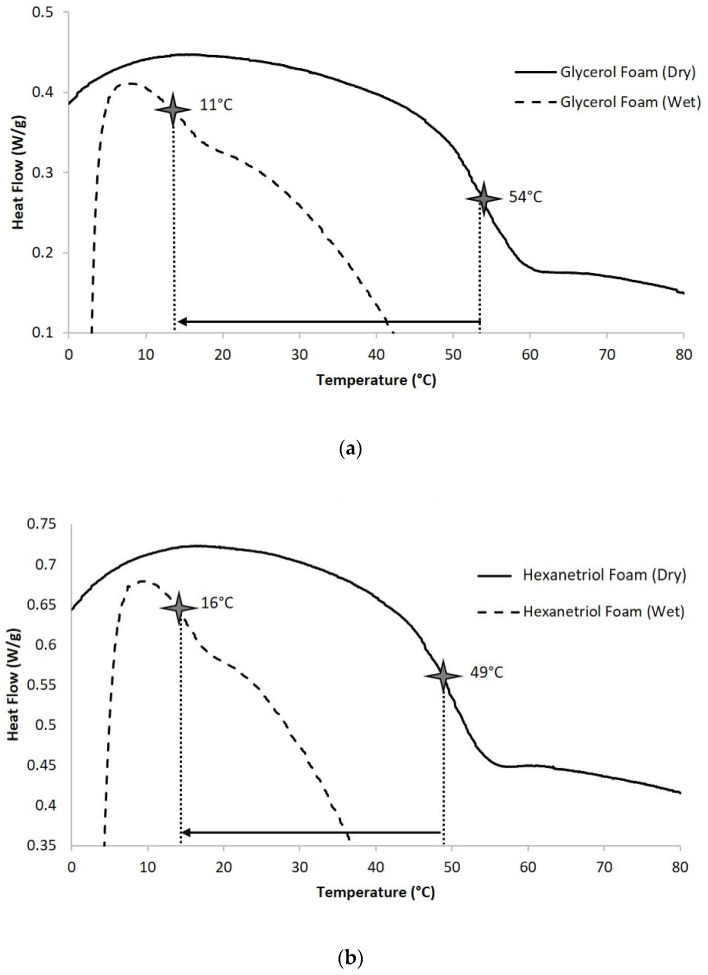
Dry and wet T_g_ comparison for glycerol foams (**a**) and hexanetriol foams (**b**).

**Figure 5 polymers-12-02290-f005:**
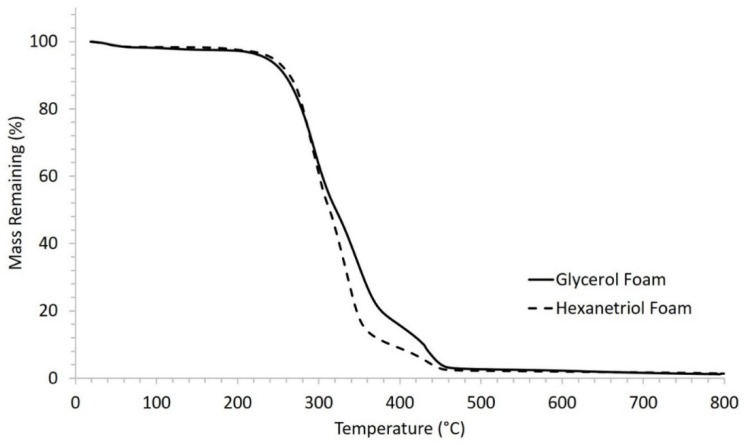
Thermal degradation profiles of glycerol and hexanetriol-based SMP foams.

**Figure 6 polymers-12-02290-f006:**
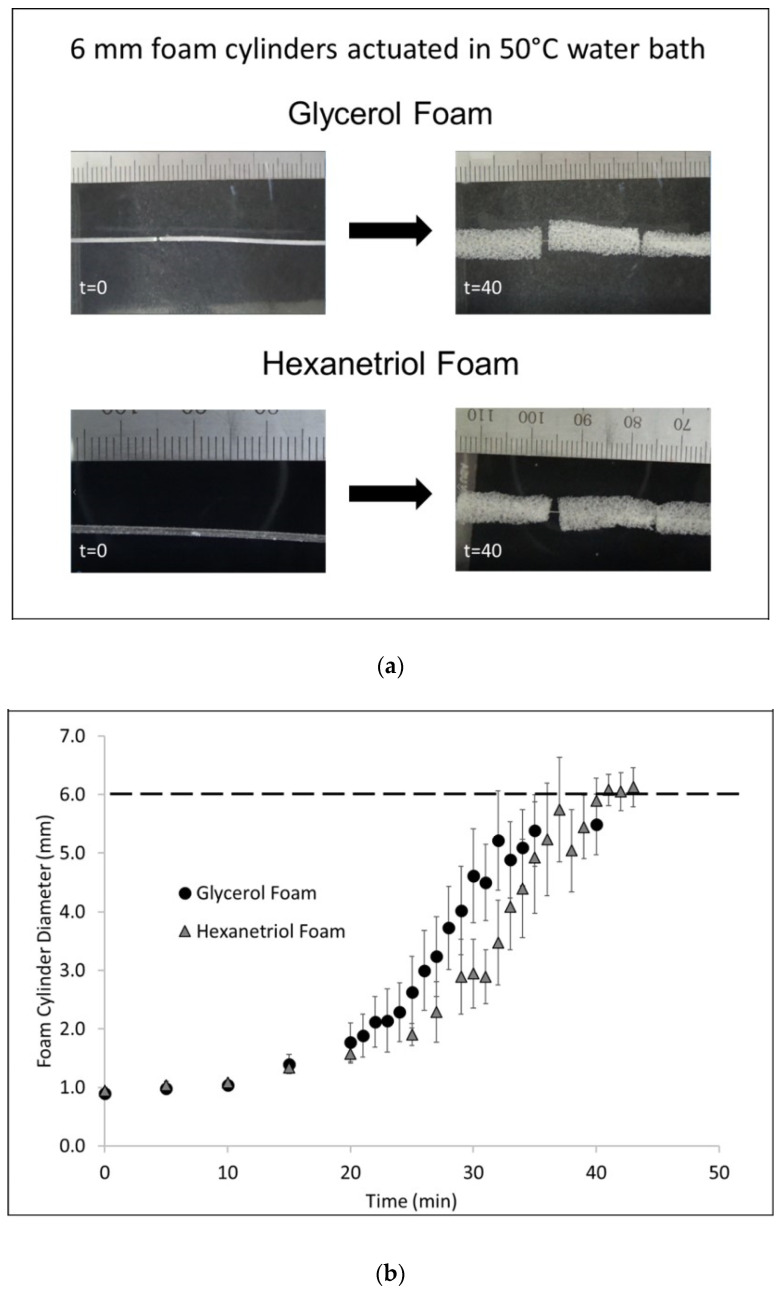
Shape memory actuation comparison of glycerol and hexanetriol SMP systems. (**a**): Images of compressed foams before actuation versus expanded foams (6 mm) post-actuation in 50 °C water. (**b**): Plot of foam cylinder diameter versus time for both foam compositions.

**Figure 7 polymers-12-02290-f007:**
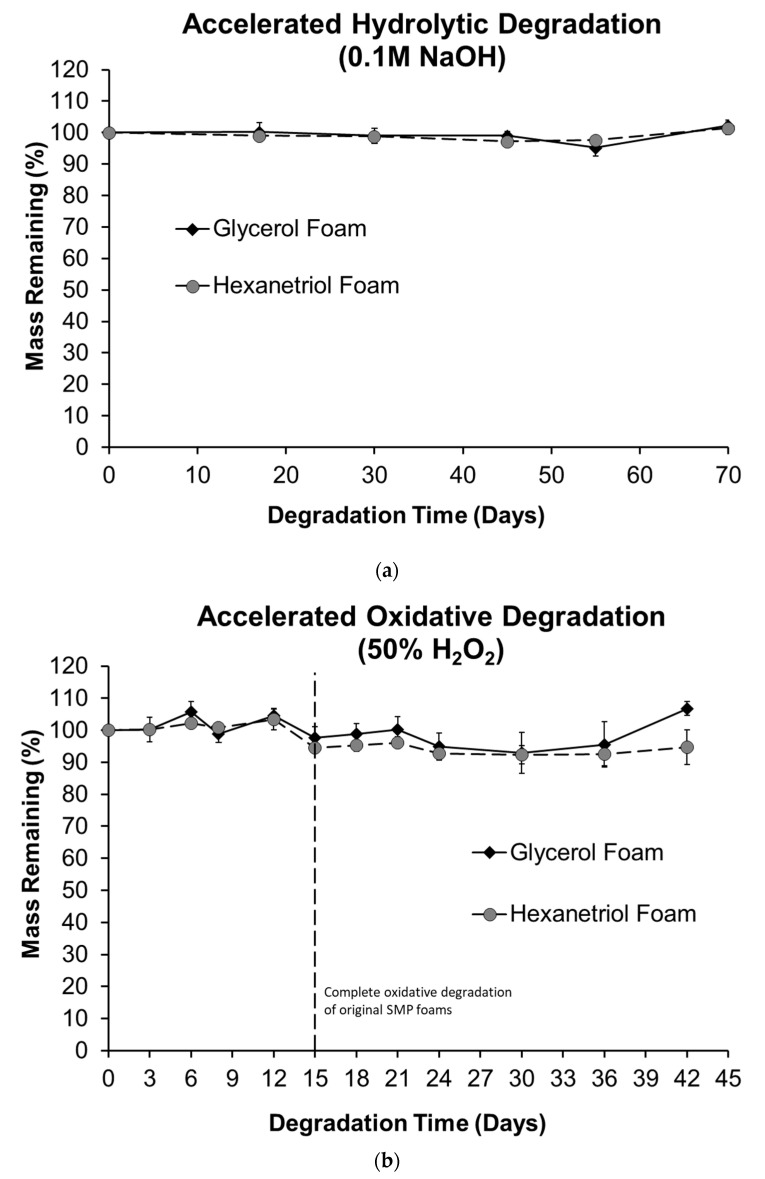
(**a**) Accelerated hydrolytic (0.1 M NaOH) degradation plots and (**b**) accelerated oxidative (50% H_2_O_2_) degradation plots of glycerol and hexanetriol SMP foams. The dashed vertical line at 15 days, for the accelerated oxidative degradation plot, indicates 100% mass loss in the original system [21].

**Table 1 polymers-12-02290-t001:** Weight percentage of all monomers used during the foaming process.

Composition	Weight Percent (wt.%)									PPH
	HDI	BEP	T-131	BL-22	DC 198	DC 5943	G	HT	Water	Enovate
Glycerol Foam	61.14	20.38	0.27	0.64	4.05	3.06	7.79	0.00	2.67	5.18
Hexanetriol Foam	57.71	19.03	2.50	0.62	3.99	2.98	0.00	10.63	2.54	5.33

**Table 2 polymers-12-02290-t002:** Summary of pore size of the shape memory polymers (SMP) foam formulations.

Composition	Axial Pore Size (µm)	Transverse Pore Size (µm)
Glycerol foam	1404 ± 121	1107 ± 69
Hexanetriol foam	957 ± 82	580 ± 47

**Table 3 polymers-12-02290-t003:** Glass Transition Temperatures (Tg) of the SMP foam formulations.

Composition	Dry T_g_ (°C)	Wet T_g_ (°C)
Glycerol foam	54.2 ± 0.7	15.6 ± 1.1
Hexanetriol foam	49.4 ± 0.3	15.5 ± 0.5

**Table 4 polymers-12-02290-t004:** Mechanical and shape memory properties of the SMP foam formulations.

Composition	Strain at Break (%)		Modulus (kPa)		R_f_ (%)	R_r_ (%)
	Dry	Water-Plasticized	Dry	Water-Plasticized		
Glycerol Foam	274 ± 41	558 ± 29	130 ± 30	8 ± 1	99.2 ± 0.2	99.3 ± 0.8
Hexanetriol Foam	247 ± 29	521 ± 43	390 ± 30	15 ± 3	98.0 ± 1.5	98.3 ± 0.3

**Table 5 polymers-12-02290-t005:** Shape recovery properties of SMP foam formulations in aqueous conditions.

Composition	Volume Recovery (%)	Volume Expansion (x)
Glycerol foam	84 ± 15	34 ± 6
Hexanetriol foam	105 ± 11	42 ± 6

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
