# Peer review of "Shape Memory Polymer Foams Synthesized Using Glycerol and Hexanetriol for Enhanced Degradation Resistance"

_polymers, 2020, doi:10.3390/polym12102290_

Round 1
Reviewer 1 Report
This manuscript describes the synthesis of two polyurethane foams and investigated their physical and chemical properties as well as their degradation profile against oxidants and basic solutions. In general, the data in this paper are promising and the manuscript is well written. It should be published in Polymers after minor revision.
Other comments:
- Since the authors claim that they have synthesized a bio-stable SMP foam, evaluation on bio-degradation should be performed and added as mentioned in the discussion section. Moreover, the study on protein absorption may also be helpful for its bio-application.
- The mechanical performance of SMP is crucial to its performance. The tensile strength before and after immersed in water should be conducted. Also, the shape fixing and recovery rate of the SMP foam is missing.
- Since the Tg of the wet SMP foam decreased below ambient temperature, why the shape memory actuation was performed at 50 °C. How about the shape memory performance of the foam at room temperature or body temperature?
- It seems to me both SMP foams exhibited similar thermal and shape memory performances even though the authors suggest they have different recovery rate and expansion. Could the authors put more comments on the difference of these two SMP foams. For example, which one is more suitable for occlusion applications as mentioned in the manuscript.
Author Response
The authors would like to express gratitude to the reviewers of Polymers for their review of our submission entitled “Shape Memory Polymer Foams Synthesized Using Glycerol and Hexanetriol for Enhanced Degradation Resistance”. We understand and appreciate the time spent on the review and sincerely appreciate the thoughtful comments and the constructive suggestions for improvement.
We have revised the manuscript as described in the response below.
Reviewer 1
Comments and Suggestions for Authors
This manuscript describes the synthesis of two polyurethane foams and investigated their physical and chemical properties as well as their degradation profile against oxidants and basic solutions. In general, the data in this paper are promising and the manuscript is well written. It should be published in Polymers after minor revision.
Other comments:
- Since the authors claim that they have synthesized a bio-stable SMP foam, evaluation on bio-degradation should be performed and added as mentioned in the discussion section. Moreover, the study on protein absorption may also be helpful for its bio-application.
The authors agree that a bio-degradation study needs to occur to achieve a full understanding of how the foam behaves in vivo. In the discussion section, it is suggested that future studies must include an in vivo degradation evaluation in an animal model. Additionally, an in vitro study may be conducted using a real-time degradation approach (PBS and 3% H2O2) as recommended by ISO10993-13: Biological evaluation of medical devices. Evaluating protein absorption onto the polymer matrix will further the understanding of potential material interactions with the in vivo environment. This is a great suggestion and we will utilize this study in the future.
Furthermore, based on this reviewer comment, we have removed claims of “biostability” from the manuscript and have recommended future studies that will be conducted to evaluate SMP biostability in PBS solutions which are more representative of an in vivo environment. This note has been made on page 15 line 376 and 392. Claims of biostability were removed from page 1 line 16 and 23, page 2 line 80 and 81, page 14 line 323 and 330, page 15 line 374, 387, and 390.
1. The mechanical performance of SMP is crucial to its performance. The tensile strength before and after immersed in water should be conducted. Also, the shape fixing and recovery rate of the SMP foam is missing.
We have conducted the recommended study and added the data to the manuscript. Tensile testing was conducted on dry and water-plasticized foams. The protocol is described on page 4 line 161 and the results are presented on page 10 line 265. Shape fixity and shape recovery were also calculated for each formulation. The protocol is described on page 5 line 174 and the results are presented on page 10 line 274.
This is a good recommendation and the authors believe that these studies add value to understanding the material properties.
2. Since the Tg of the wet SMP foam decreased below ambient temperature, why the shape memory actuation was performed at 50 °C. How about the shape memory performance of the foam at room temperature or body temperature?
Actuation studies were conducted at 50°C to show shape recovery of the foam in an aqueous environment. While the foam still actuates at 37°C and room temperature water, the recovery profiles are elongated (hours instead of minutes). Since the polymer network is inherently more hydrophobic than systems designed by Singhal et al. (2012), water penetration is slow and additional heat is necessary to drive material actuation. A slow expansion profile, at 37°C, would be well-suited for a slow release drug profile application, however for the purposes of an initial material characterization study, we selected 50°C water to evaluate volume recovery and expansion.
3. It seems to me both SMP foams exhibited similar thermal and shape memory performances even though the authors suggest they have different recovery rate and expansion. Could the authors put more comments on the difference of these two SMP foams. For example, which one is more suitable for occlusion applications as mentioned in the manuscript.
While the thermal and shape memory properties are close, both foams exhibit slightly different actuation profiles. Figure 6 shows that the recovery profile for the glycerol foam starts to plateau after 30 minutes and at 40 minutes, the foam has achieved maximum recovery. The hexanetriol foam continues to expand at 35 minutes and starts to plateau at 40 minutes. A comment was added to page 14, line 357. Furthermore, Table 4 reported on the volume recovery of both foams compared to their original respective cylinder diameters. The glycerol foam recovered up to 84%, on average, while the hexanetriol foam had a volume recovery of 105% indicating that the foam expanded past its original diameter. This is an interesting property because not only did the polymer achieve full volume recovery to its original size, it also swelled slightly beyond its original shape likely due to the foam struts expanding from water plasticization.
In terms of a practical application, both systems will require more fine-tuning to adjust the pore sizes and the dry glass transition temperatures. Typically, for neurological occlusion applications, smaller pores are ideal to allow for trans-catheter access and appropriate volume filling. Higher glass transition temperatures are appropriate for this application due to the longer working time requirements by the physician. For peripheral applications, the pore sizes of these foams are acceptable, however the glass transition temperature should be lowered to allow for tunable recovery and potentially shorter actuation times. Based on the results of the current study, the hexanetriol foam will be a better candidate for optimization. The full volume recovery of the hexanetriol foam, in an aqueous environment, is promising along with a lower dry glass transition temperature which will result in faster actuation.
Reviewer 2 Report
The authors investigated shape-memory polymers based on polyurethanes. For this purpose, polymer networks were synthesized and studied regarding their structure as well as their SMP-behavior. Finally, degradation tests were performed. The study is in general performed in an apprioate manner, however, some major points have to be addressed before publishing. These are listed in the following:
- There are many reports out describing SMP based on polyurethanes, also for biomedical application. What is the major improvement of your system?
- Did you change the ratios of the different compoundsin order to find the optimal composition for the application?
- On page 6, discussion IR-spectra: Why do you mention ureas? You just react alcohols and isocyanates resulting in urethanes. What is the origin of the ureas? Did you use an amine also?
- Furthermore, the urethane moiety itself did not contain a secondary amine. It is an amide of the carbonic acid.
- Did you anneal the polymers prior to usage? Will it change the properties of the materials?
- The DSC-diagrams could be presented in a better manner.
- Did you perform swelling tests of your material? How much water can be taken up and to what about the volume expansion? In principle you should test it for water and PBS-buffer.
- You should quantify your shape-memory behavior by rheology measurements in order to measure the fixitiy and recovery rate.
- Is the shape-memory behavior based on the temperature or on the water or on both? In principle you did not reach the Tg of the glycerol form; therefore, the SMP should be based on water or the coombination of both. However, you should perform SMP tests just at 50 °C (without water) and with water at room temperature. Furthermore, you should determine the thermal properties of the swollen polymer (Tg and TGA).
- The degradation studies are not performed in a suitable manner. Just measuring the mass is not sufficient. You should also look at the chemical structure by measuring IR before and after (at least). Furthermore, the SMP-behavior could be tested after degradation tests (swelling would also be a possibility)
Author Response
The authors would like to express gratitude to the reviewers of Polymers for their review of our submission entitled “Shape Memory Polymer Foams Synthesized Using Glycerol and Hexanetriol for Enhanced Degradation Resistance”. We understand and appreciate the time spent on the review and sincerely appreciate the thoughtful comments and the constructive suggestions for improvement.
We have revised the manuscript as described in the response below.
Reviewer 2
Comments and Suggestions for Authors
The authors investigated shape-memory polymers based on polyurethanes. For this purpose, polymer networks were synthesized and studied regarding their structure as well as their SMP-behavior. Finally, degradation tests were performed. The study is in general performed in an apprioate manner, however, some major points have to be addressed before publishing. These are listed in the following:
1. There are many reports out describing SMP based on polyurethanes, also for biomedical application. What is the major improvement of your system?
While there are several polymer systems reported in the literature with various properties, relevant to their intended application, we focused on developing a low-density, high-porosity, amorphous material with a non-degradable chemistry. One of the unique properties of our system is the lack of degradable functional groups within the polymer network. The starting materials were selected explicitly to eliminate ethers, esters, and amine groups that are easily degraded via oxidative and hydrolytic mechanisms. There are several polyurethane systems described in the literature, specifically for biomedical applications, however we believe that our system is unique in its lack of degradation via oxidation and hydrolysis. Furthermore, our system has shape memory properties like the polymers developed by Singhal et al. while also maintaining tunable thermal and actuation profiles. Our system provides an additional level of tunability by using degradation-resistant starting materials during foam fabrication. The unique properties of the material system are described on Page 2, Line 55 of the revised manuscript.
2. Did you change the ratios of the different compoundsin order to find the optimal composition for the application?
The authors employed different molar ratios for the compounds used in this study and well as studied different compound blends to fabricate these materials. However, compound miscibility and foamability served as a significant challenge for this work and two viable foam formulations were selected for further characterization.
3. On page 6, discussion IR-spectra: Why do you mention ureas? You just react alcohols and isocyanates resulting in urethanes. What is the origin of the ureas? Did you use an amine also?
The ureas are a result of a reaction caused by the chemical blowing agent, water, which is used during foam fabrication. Water molecules react with isocyanate groups to generate carbon dioxide which caused bubble formation. Another byproduct of this reaction is primary amines which can further react with the isocyanates present in the reaction vessel to produce urea linkages. These urea linkages become a part of the polyurethane polymer network, thereby resulting in a polyurethane-urea network. This note as been made on page 7 line 246 of the manuscript.
4. Furthermore, the urethane moiety itself did not contain a secondary amine. It is an amide of the carbonic acid.
We agree that the source of the amine is not the amide group on the urethane functionality. Amines are a byproduct of the water-isocyanate reaction which further react with free isocyanates to generate ureas.
5. Did you anneal the polymers prior to usage? Will it change the properties of the materials?
The foams were heated above their Tg and cooled down to room temperature prior to degradation testing and expansion studies. This annealing step is conducted to remove residual stress in the material from the foam blowing process. As with most polymer systems, annealing the material helps improve mechanical properties while also providing consistency from batch to batch. Our polymer system is thermoset therefore some of the larger benefits of annealing will not be represented, such as the melting of irregular crystalline sections to generate better formed ones. However, the overall goal of removing inherent stress from the material is still achieved during this step. Furthermore, glass transition temperature was acquired using a heat/cool/heat cycle there the first heat cycle was used to anneal the polymer sample and remove any volatiles and stresses present in the material. Mechanical testing was also conducted on samples that were annealed for 30 minutes and allowed to cool to room temperature, prior to tensile testing. The annealing process is added to the Methods section on page 3 line 109, page 4 line 168, and page 5 line 176.
6. The DSC-diagrams could be presented in a better manner.
The DSC diagrams have been updated (Figure 4)
7. Did you perform swelling tests of your material? How much water can be taken up and to what about the volume expansion? In principle you should test it for water and PBS-buffer.
Volume expansion and volume recovery studies were conducted per the protocol described on page 4 of the manuscript. Our material is a shape memory polymer (SMP) with heat being the primary actuator. Additionally, moisture plasticization lowers the glass transition temperature of the polymer by disrupting hydrogen bonding and thereby aids in polymer actuation as well. In the volume expansion study, we actuated the SMP in water and observed properties such as volume expansion and recovery. Both SMP foams had a volume recovery greater than 80% and volume expansion of 34x and 42x for the glycerol foam and hexanetriol foam, respectively. Only hexanetriol foam appeared to swell in water, to a diameter larger than its original, as the volume recovery was 105%. Since our material is a hydrophobic SMP, swelling is not an expected characteristic, however some swelling was observed in the hexanetriol foam only.
8. You should quantify your shape-memory behavior by rheology measurements in order to measure the fixitiy and recovery rate.
Shape fixity and shape recovery of both foam formulations was evaluated per the reviewer’s request. The protocol for this testing is described on page 5 line 174 and the results are presented on page 10 line 272. This is a great recommendation and the results of this experiment will aid with a better understanding of the material systems.
9. Is the shape-memory behavior based on the temperature or on the water or on both? In principle you did not reach the Tg of the glycerol form; therefore, the SMP should be based on water or the coombination of both. However, you should perform SMP tests just at 50 °C (without water) and with water at room temperature. Furthermore, you should determine the thermal properties of the swollen polymer (Tg and TGA).
The shape memory profile of our polymer system is temperature-driven where the material achieves shape recovery once it is heated above its glass transition temperature. Water plasticization is a phenomenon that aids shape recovery by lowering the polymer’s Tg via a disruption in hydrogen bonding between the urethane and urea groups. Water molecules penetrate the polymer network and break the hydrogen bonds within adjacent urethane/urea functionalities. Water molecules then engage in hydrogen bonding with the urethane/urea groups instead which in turn drives down the polymer Tg. The effects of water plasticization are described on Page 12, Lines 284-292.
Expansion testing in a dry environment, at 50°C, would yield a very slow expansion profile. The dry Tg of both foam formulations are close to 50°C which indicates that the material would expand, however the transition would be slow (over several hours). Typically, for shape recovery in dry conditions, the polymer is heated to at least 30°C above its Tg to observe significant actuation (Singhal et al., 2012).
Similarly, foam expansion in 37°C water would be slow due to the hydrophobic nature of the polymer network. The starting materials selected to develop a non-degradable foam do not contain hydrophilic functional groups such as amines, ethers, and esters. The lack of these functional groups prevents material degradation while also making the polymer network hydrophobic compared to previously designed systems by Singhal et al which used polyols containing amine groups. For the purposes of this initial material characterization study, we utilized 50°C water to demonstrate the material’s shape memory behavior. While the SMP foam would still expand at 37°C water, the actuation profile will be elongated due to slow moisture plasticization of the polymer network.
Thermal properties of the wet polymer were measured via DSC, per the protocol described on Page 4, Line 144 of the manuscript.
10. The degradation studies are not performed in a suitable manner. Just measuring the mass is not sufficient. You should also look at the chemical structure by measuring IR before and after (at least). Furthermore, the SMP-behavior could be tested after degradation tests (swelling would also be a possibility)
The authors believe that since this study was primarily a material characterization study to evaluate shape memory polymer properties and degradation stability, the mass loss study was sufficient for screening viable chemistries that can be further fine-tuned and studied. We agree with the reviewer that a more detailed degradation study would be necessary to fully evaluate changes to the thermo-mechanical properties and chemical structure of the polymer during and after exposure to the various degradation solutions. However, this study would be more appropriate once a foam formulation is modified for a specific biological application and foam morphology, chemical composition, and thermal properties are fixed. The formulations presented in this study are novel chemistries developed with a potential for use in biological applications and would require further tuning of pore morphology and thermal properties before they are suitable for a medical application.
We propose to conduct a detailed degradation study with full a chemical evaluation of the SMP foam along with an analysis of shape memory properties after removal from the degradation solution, like the study conducted by Weems et al. (2017, Acta Biomaterialia). In this study, the authors evaluated the degradation mechanism, chemical changes, and cytocompatibility of the degraded SMPs which had previously shown efficacy for use in specific embolic devices per the studies published by Landsman et al. (2016, Journal of the Mechanical Behavior of Biomedical Materials) and Boyle et al. (2015, Journal of Biomedical Materials Research B: Applied Biomaterials). A comprehensive degradation study has been recommended on page 15 line 376, per the reviewer comment.
Round 2
Reviewer 2 Report
The authors changed the manuscript in a sufficient manner. However, I still have problems with "non-degradable" SMPs. Just measuring the mass does not tell anyhting about the structure of the polymers. Urethanes can be reversible as well (it is not accpeted in the current setup), however, measuring an IR before and after heating would proof the absence of changes of the chemical structure in a very simple manner.
Author Response
The authors would like to thanks the reviewers of Polymers for their review of our submission entitled “Shape Memory Polymer Foams Synthesized Using Glycerol and Hexanetriol for Enhanced Degradation Resistance”. We appreciate the re-evaluation of the manuscript and the constructive suggestions for improvement.
We have revised the manuscript as described in the response below.
Reviewer 2
The authors changed the manuscript in a sufficient manner. However, I still have problems with "non-degradable" SMPs. Just measuring the mass does not tell anyhting about the structure of the polymers. Urethanes can be reversible as well (it is not accpeted in the current setup), however, measuring an IR before and after heating would proof the absence of changes of the chemical structure in a very simple manner.
All language stating “non-degradable” has been replaced with “degradation-resistant” on page 14 line 333 and page 15 line 377, 390, and 393. Additionally, samples from the glycerol foam and the hexanetriol foam were heated at 150°C for 30 minutes. FTIR spectra, in absorbance mode, were collected for each sample before and after heating to evaluate changes to the chemical spectra, especially the urethane linkages, due to heat. TGA thermogram on page 10 of the manuscript indicates that the onset of thermal degradation of the SMP foam is at approximately 200°C, therefore 150°C was selected as the temperature to which we heated the samples to ensure that thermal degradation did not contribute to chemical changes. FTIR spectra, shown in the pages below, indicate no changes to chemical peaks for both foam formulations after heating. The spectra for non-heated and heated foam overlay well with no new peaks or shifts in existing peaks. Furthermore, no changes can be observed to the urethane and urea carbonyl peaks at 1689 cm-1 and 1646 cm-1, respectively, after the samples were heated. This study indicates that the urethane and urea groups are stable under heat and the urethane linkages are irreversible. A note pertaining to the thermal stability of these materials, as verified by FTIR, was added to page 9 line 265